# Regulations on the Use of Antibiotics in Livestock Production in South America: A Comparative Literature Analysis

**DOI:** 10.3390/antibiotics12081303

**Published:** 2023-08-09

**Authors:** Rafael Almeida Da Silva, Nelson Enrique Arenas, Vera Lucia Luiza, Jorge Antonio Zepeda Bermudez, Sian E. Clarke

**Affiliations:** 1Pharmaceutical Assistance Department, Sergio Arouca National School of Public Health, Oswaldo Cruz Foundation, Rio de Janeiro 21040-361, Brazil; vera.luiza@fiocruz.br (V.L.L.); jorge.bermudez@fiocruz.br (J.A.Z.B.); 2Faculty of Sciences, Universidad Antonio Nariño, Bogotá 110231, Colombia; nearenass@unal.edu.co; 3Disease Control Department, Faculty of Infectious and Tropical Diseases, London School of Hygiene & Tropical Medicine, London WC1E 7HT, UK; sian.clarke@lshtm.ac.uk

**Keywords:** drug resistance, microbial, One Health, livestock industry, South America, legislation

## Abstract

As a global health problem, antimicrobial resistance (AMR) crosses national borders, leading UN (United Nations) multilateral agencies to call for all countries to improve the stewardship of antibiotics in humans and animals. South American countries have changed their regulations regarding antibiotic use in livestock production. This literature review examines how far the five largest meat-producing countries in South America (Argentina, Brazil, Chile, Colombia, and Uruguay) have come in terms of the relevant legislation. Rules on market entry (marketing authorization and official distribution systems) are already set in all countries examined. Four countries do not allow growth promoters based on critically important antibiotics, and countries have also begun to set guidelines and minimum welfare and biosecurity requirements to reduce the therapeutic demand for antibiotics. Nonetheless, there are aspects related to the distribution, use, and disposal of antibiotics that need to be developed further. In conclusion, legislation in South American countries is moving towards the goals set by UN multilateral agencies, but more can be done. Differences between countries’ rules and the gold standards set by the World Organization for Animal Health (OIE), World Health Organization (WHO), and Food and Agriculture Organization of the United Nations (FAO) reveal possible adaptations to the countries’ realities. Further studies must examine compliance with the legislation already set and investigate other tools that can be used alongside legislation as a driving force to change stakeholder behaviour.

## 1. Introduction

Antimicrobial resistance (AMR) is a growing, cumulative global health challenge. Globally, an estimated 700,000 people die each year from antimicrobial-resistant infectious diseases [1]. If the problem continues to be neglected or if there are no control strategies, antibiotics will become increasingly ineffective, and AMR could cause 10 million deaths per year by 2050, jeopardizing a century of progress in human health [1].

AMR bacteria occur as a result of bacterial plasticity towards adaptation in response to environmental pressure by antibiotic compounds [2]. This complex problem is associated with several factors including the use of antibiotics in livestock production to ensure animal welfare, health, and human food security when either excessive or insufficient doses can increase AMR bacteria in animals and humans. Dissemination of AMR bacteria between animals, humans, and the environment occurs through animal contact, meat consumption, and disposal of livestock solid waste in the environment [3].

Since 1998, antibiotic stewardship plans to optimise antibiotic use in livestock production have been discussed at international forums such as the annual World Health Assembly (WHA) [4]. The strategy to mitigate AMR bacteria is based on the One Health approach, which was defined by consensus as “a collaborative, multisectoral, and transdisciplinary approach working at the local, national, regional, and global levels […] recognizing the interconnection between people, animals, plants, and their shared environment” [5]. By 2022, 86 countries had approved national action plans on AMR, representing 44% of the 194 countries in the world [6].

As a global health problem, AMR crosses national borders, and no country on its own could contain its advance [7]. Therefore, it is important to strengthen and align country strategies to contain AMR bacteria with the global action plan and international standards recommended by the World Health Organization (WHO), Food and Agriculture Organization of the United Nations (FAO), and the World Organization for Animal Health (OIE) [8].

South America is the fourth largest meat-producing region in the world. Between 1961 and 2018, it increased its production by 608%, reaching 46.12 million tons of meat produced in 2018 [9]. In addition, it is the third largest region in terms of meat consumption. In 2013, it was estimated that this region consumed 81.49 kg/person/year.

Whereas some of Brazil’s trade partners for meat have already improved their legislation on antibiotic use in animals—such as the European Union, which banned the use of growth promotion in 2006 [10]—it is not certain if South America is heading in the same direction. Thus, a comparison between the legislation of South American countries and international standards is important from a public health as well as from an economic perspective. This work could lead to a better understanding of where and how to improve policies in food safety to secure global public health and the international livestock market.

Our study presents findings from comparative analyses of local policies, guidelines, and official programs on antibiotic use and AMR bacteria in livestock production in five selected South American countries, in relation to the standards established by the FAO, WHO [11], and OIE [12], to determine: (i) how far South American countries have come on legislation about antibiotic use in livestock, and (ii) whether these legislative changes are moving in the same direction as the goals established by UN multilateral agencies.

## 2. Results

### 2.1. Data Screening and Overview

A total of 77 documents were analysed, including normative and national rules, examining each successive step in the antibiotic supply chain between producer and consumer (Figure 1). In most of the countries evaluated, institutions related to food production (from both animal and plant sources) were responsible for establishing rules on the use of medicine in animals and plant production, from marketing authorization to pharmacovigilance. Colombia was the only country with the One Health approach that shared responsibilities between the Ministry of Health and institutions connected to it, through the Instituto Nacional de Vigilancia de Medicamentos e Alimentos (INVIMA).

A summary (Figure 1) was created to show which points in the antibiotic supply chain are supervised by the authorities (1) or prescribers (2).
Responsibility of the government to supervise or conduct monitoring and vigilance.Responsibility of the veterinarian to supervise.

The level of development for each country regarding their legislation on antibiotic supply and use is synthesised in Table 1, together with the guidance issued by relevant UN agencies. Overall, Chile was classified as strong in a greater number of categories than the other countries, while Argentina was the weakest.

### 2.2. Marketing Authorization (Manufacture and Import)

All countries are classified as strong in this category since they have a marketing authorization system to register veterinary medical products (VMPs) containing antibiotics and growth promoters (in countries where the latter are allowed); to authorize the manufacture of these veterinary products by the veterinary pharmaceutical industry; and to grant animal feed manufacturers licences to mix antibiotics in the feed [13,14,15,16,17,18,19,20].

The criteria to approve the registration of veterinary products in all countries are quality, safety, efficacy, and withdrawal period [13,14,16,21,22,23]. Maximum residue limit was mentioned as an additional criterion in the legislation in Argentina [24], Brazil [14], Chile [16], and Uruguay [25]. However, only Argentina [24], Brazil [26], and Uruguay [25] included acceptable daily intake as a requirement to register veterinary products.

All countries required the veterinary pharmaceutical industry to comply with the provisions of good manufacturing practices [13,14,16,21,23,27]. Additionally, Chile [16] and Colombia [17] required the industry to have quality control laboratories following good laboratory practices, but only Colombia demanded good clinical practices following the International Cooperation on Harmonisation of Technique for Registration of Veterinary Medical Products [18]. All animal feed manufacturers must follow good manufacturing practices established by the authorities in order to be registered [28,29,30,31,32,33].

### 2.3. Container Labelling and Advertising

In this category, all countries were classified as “intermediate” since they included most of the information described in the OIE gold standard for the labelling of VMPs containing antibiotics and growth promoters. In Argentina [13,28] and Colombia [17,30], the legislation missed “appropriate animal age” on the labelling for VMPs containing antibiotics, whilst Brazil [14,34], Chile [16,35], and Uruguay [19,36] did not include the animal production category (dairy or beef). Regulations in Chile did not mention warnings, cautions, and indications of use in antibiotic-containing feed [35]. Guidance on labelling in Argentina specified that antibiotics must not be used as a growth promoter in food-producing animals whose products or by-products will be exported to the European Union (EU) or other countries with the same requirement [37], but this did not extend to the domestic market.

None of the countries had regulations forbidding the advertising of VMPs and growth promoters aimed at food animal producers or financial incentives to prescribers and suppliers. Only Colombian legislation limits advertisement, which is allowed in scientific journals or technical publications [38,39].

### 2.4. Distribution

In all countries, retail distributors and animal feed manufacturers must be authorised by a government authority to market VMPs containing antibiotics or growth promoters. To market these products, they must guarantee the storage conditions mentioned on the labelling (temperature, humidity, light, ventilation, etc.) established by the manufacturer to preserve the characteristics (quality and efficacy) and avoid contamination or other damage to VMPs containing antibiotics and growth promoters [13,14,16,18,19,20,28,29,30,35,40,41,42]. Also, animal feed manufacturers must mix the feed only with antibiotics approved by the governmental authority, and follow instructions established in the drug premix label and veterinarian indications.

Colombia and Uruguay achieved a higher standard than other countries since they not only required a veterinarian’s prescription as a condition of supplying VMPs containing antibiotics but also required both retail distributors and animal feed manufacturers to keep a record of the prescriptions [42,43,44]. Regulations also stipulated that suppliers must follow instructions issued by the veterinary pharmaceutical industry to dispose of expired products [19,36,45]. Argentina, Brazil, and Chile, however, did not meet all the criteria for prescription and reporting established by the gold standard [12], and were therefore classified as “intermediate”. Animal feed manufacturers and retail distributors in Argentina [13] and Chile [46] can only supply VMPs containing antibiotics with a veterinarian’s prescription. However, in Brazil, retail distributors can supply VMPs containing antibiotics (injectable or orally administered in water) without a veterinarian’s prescription, since antibiotics are not considered controlled substances [47]. Also, in Brazil [29] and Argentina [28], only animal feed manufacturers are required to retain the veterinarian’s prescription to keep a record of the antibiotics supplied. Regulation in Chile was slightly more advanced, with all retail distributors required to keep records, but only for the third- and fourth-generation classes of fluoroquinolone and cephalosporins [46]. Animal feed manufacturers in Chile must also submit all information about the quantity of feed with antibiotics produced, per species and production stage, to the government authority [35]. None of the five countries examined required a veterinarian’s prescription to supply growth promoters based on an antibiotic.

The international gold standard does not specify the professional background of the technical manager responsible for the suppliers, nor the demand to get access to growth promoters, and was thus not taken into consideration in scoring. Nonetheless, it is worth noting how the required background of technical managers differs from country to country. For retail distributors, Argentina [13], Brazil [14], and Uruguay only allow [19,20] veterinarians as technical managers, while Chile allowed veterinarians and pharmacists [16].

For animal feed manufacturers, Chile only allows veterinarians and other professionals with knowledge of animal health and food safety [35]; Uruguay [44], Argentina [28], and Brazil [29] allow both veterinarians and agronomists as technical managers, though Argentina mentions that other professionals with similar careers can also take the position. Brazil [29] also allows zootechnicians to perform the same function.

### 2.5. Prescription

All countries specify veterinarians as the only professionals allowed to prescribe antibiotics to animals in their legislation [13,42,43,48,49]. Brazil and Chile were classified as “strong” for different reasons. Brazil has defined all the topics to guide prescription (clinical examination, experience, and diagnostic); moreover, it has a priority scale for use based on the OIE and WHO list of important antibiotics for human and veterinary medicine [50]. Although no similar guideline was found in Chile, restrictions and conditions have been established by resolutions [51,52] to the prescriber, and the professional ethical code [53] mentions that veterinarians should take care when prescribing, using, administering, and disposing of VMPs containing antibiotics, considering the damage that could occur to the environment and public health, and that veterinarians should teach users about AMR bacteria and the rational use of antibiotics.

Colombia and Uruguay were classified as “intermediate”. The national guidelines [45,54] indicated that prescription should be based on clinical examination, experience, and diagnosis, but there was no priority scale for use based on the OIE and WHO list in these documents. These countries have specified that, whenever possible, VMPs containing antibiotics should be administered under veterinary supervision. In cases when this is not possible, Uruguay mentions that the veterinarian should give clear instructions, including the dosage, route of administration, and withdrawal period [54]. Argentina was classified as “weak” since no guidelines for responsible, prudent use of VMPs containing antibiotics and restrictions or conditions of administering antibiotics in livestock (cattle, pigs, and chicken) were found.

### 2.6. Use of Veterinary Medical Products Containing Antibiotics

International guidelines for antibiotic stewardship of VMPs distinguish between three purposes of use (therapeutic, metaphylactic, and preventive), referring to all routes of administration (injectable or orally administered through feed or water) [12].

#### 2.6.1. Therapeutic and Metaphylactic Use

Only Brazil [55] and Chile [52] stated the classes of the third and fourth generation of cephalosporins, fluoroquinolone, and polymyxins (colistin) as medicines of last resort for these two purposes of use. Colombia [45] and Uruguay [54] were “intermediate” because there was no mention of these restrictions imposed by WHO and OIE in the official guidelines. Argentina was the only country classified as “weak” due to the lack of guidelines or specific rules. We only found veterinary guidelines published by the Spanish Government that were used as a reference on an unofficial site to support pig and cattle producers: (https://www.produccion-animal.com.ar/ (accessed on 15 October 2022)). Nonetheless, Argentina was the only country to ban colistin for all purposes of use [56].

#### 2.6.2. Preventive Use

Guidelines on preventive use were more limited, and none of the five countries met the international standard. Chile [46] and Brazil [55] affirmed that the preventive use of VMPs containing antibiotics was permissible in specific cases when it is deemed necessary, although rules in Chile [46] further stated that in these cases antibiotics must be administered to a single animal or a limited number of animals. Guidelines in Uruguay [54] and Colombia [45] did not indicate restrictions on which antibiotics could be used for this purpose, while for Argentina, no relevant legislation was found.

#### 2.6.3. Non-Veterinary Medical Use (Growth Promotion)

International guidelines regarding the administration of antibiotics to animals for any purpose other than the treatment, control, or prevention of infectious diseases include growth promotion, which is intended to increase the rate of weight gain or the efficiency of feed utilisation [12]. In this respect, Argentina, Chile, and Colombia were all classified as “strong”, while Uruguay and Brazil were “intermediate”. Growth promoters based on human therapeutic antibiotics are banned in Colombia [57], which also prohibits the use of any veterinary therapeutic antibiotics as growth promoters in dairy cattle and buffalo [40]. Chile has banned the use of growth promoters based on all classes of antibiotics for all species and production categories [46]. Although Argentina only allows certain antibiotics to be mixed with feed, namely lasalocid, maduramicin, monensin, narasin, nicarbazin, robenidine, salinomycin, and semduramicin, none of these substances [58] are considered critically important antimicrobials for human medicine by the WHO [59]. We considered this rule an indirect way to regulate both the use of antibiotics as growth promoters and VMPs containing antibiotics.

Over time, Brazil has similarly been phasing out many classes of antibiotics listed in the WHO category of Highest Priority Critically Important Antimicrobials [60,61,62,63,64]. The only antibiotic-based growth promoters currently registered are avilamycin, bacitracin, enramycin, flavomycin, lasalocid, narasin, salinomycin, sodium monensin, zinc bacitracin, and virginiamycin [65]; only bacitracin and virginiamycin are on the WHO’s list of Critically Important Antimicrobials for Human Medicine [59]. By contrast, Uruguay forbids the use of antibiotics as growth promoters in bovines and sheep [66] but no legislation regarding the use of growth promoters in pork and chicken was found.

### 2.7. Off-Label Use

Chile had the most advanced regulations, with guidelines stipulating the reasons, conditions, and restrictions for off-label use of VMPs containing antibiotics. This use can only be considered under the following conditions: there are no antibiotics registered to treat the diagnosed disease; the registered antibiotic is unavailable on the market; the dosage, period, and route of administration were not sufficient to achieve the expected results; the use of antibiotics is ineffective in treating the disease, according to the container labelling [66]. If one of these reasons is present, all subsequent mandatory conditions have to be followed to prescribe off-label use: (1) the animal must be suffering, with a risk of imminent death; (2) the animal must be under veterinarian supervision; (3) the veterinarian must know the disease he wants to treat; and (4) the veterinarian must be responsible for any collateral or unexpected effects from the prescribed antibiotic. All the above conditions must be registered, including the animal’s health history and information about the responsible veterinarian [67]. Furthermore, all information regarding the diagnosis, prescription, evolution of the treatment, and results must be recorded for two years; a long-enough withdrawal period must be established and followed to ensure that residues do not exceed the maximum residue limit (MRL); and animals, products, or by-products should not be intended for human consumption. When there is no certainty that the residue is lower than the MRL, the veterinarian must establish measures to identify and track animals, products, and by-products intended for human consumption, and discontinue use if there are side effects (local or systemic) or if the expected results are not achieved [67]. Off-label use of VMPs containing antibiotics cannot be prescribed for diseases under an official disease control and eradication program, and they must not be used as a growth promoter, or when anyone could determine the residues from the active ingredient, metabolite, or other related substances [67].

By contrast, Argentina [68], Brazil [48], Colombia [69], and Uruguay [70] simply mention in their ethical code that veterinarians are responsible for defining the conditions of responsible use in cases where there is a need for off-label use.

### 2.8. Food Animal Production

In the regulation of antibiotic use by food animal producers, Argentina, Chile, Uruguay, and Colombia were classified as “strong”, and Brazil as “intermediate”. Chile is the only country that demands that records of VMPs containing antibiotics used in production are kept by all kinds of food producers (chicken, pork, and cattle). Furthermore, twice a year producers must send information to Chile’s authorities about the biomass, consumption of colistin, fluoroquinolones, third and fourth generation of cephalosporins, and specify in which categories of animal production the substances were administered [71]. In Argentina [72] and Colombia [73], only pork and beef producers are required to keep records, and in Uruguay only beef and milk producers [74]. In contrast with other countries, Brazil does not demand that producers keep records of the veterinary products used in animal production.

Chile was the only country to require by law that food producers present an action plan with infection prevention and correction measures to reduce the need for the use VMPs containing antibiotics in animal production. Also, recognition measures are offered to producers that follow the guidelines established by the program [71]. Some progress is also seen in other countries. Argentina [75] has established a minimum requirement that all food animal producers must follow regarding animal welfare, while Uruguay [76] and Argentina [77,78] have implemented specific rules on animal welfare and biosecurity measures for chicken producers. Colombia defines some general aspects to promote animal welfare and has set minimum requirements regarding animal welfare and biosecurity for the production of cattle, buffalo [73], and pigs [79]. Brazil has a normative guidance on animal welfare in swine production but no minimum requirement on biosecurity measures was found [80]. In addition to the rules, guidelines on animal welfare were found in Argentina [81,82], Brazil [83], Chile [84], and Uruguay [85,86].

### 2.9. Monitoring and Surveillance of Antibiotic Consumption, Residues in Food, and AMR Bacteria

Brazil, Argentina, and Colombia were classified as “strong” since they have established surveillance systems for antibiotic consumption and AMR bacteria, whereas Chile only has a surveillance system for antibiotic consumption. Uruguay lacked both forms of surveillance and was thus defined as “weak” in this category. In Brazil and Chile, monitoring systems for antibiotic consumption are based on sales data. Brazil’s Agromonitora system requires that once a year the veterinary pharmaceutical industry must report sales data of all VMPs containing antibiotics (therapeutic purpose) and growth promoters. If possible, the industry should also specify the species and route of administration [87]. Despite Brazil’s progress in monitoring the use of antibiotics, no results have been published yet. A similar program has been adopted in Chile, even though only a specific list of antibiotic classes is monitored. Also, the industry must specify the route of administration, species, and purpose of use (therapeutic or growth promoter). Data on antibiotic consumption in Chile from 2014–2020 can be found on the SAG official website [88]. Uruguay has adopted a veterinary product sales control but has not included antibiotics yet. Currently, the industry, retail distributors, and others only send data on products used for tick and fly control [89]. Argentina [90] and Colombia [91] mention evaluating antimicrobial use (AMU) as part of their surveillance of antimicrobial resistance programs, but no specific source documents were found.

In terms of surveillance of AMR bacteria, Argentina [90], Brazil [92], and Colombia [91] have all established national surveillance programs, though the specified source samples vary, ranging from animal faeces, live animals, animal carcasses at slaughter, to food from an animal source (e.g., meat). None of the countries mentioned using a sentinel surveillance strategy in their program. Chile has a goal to create an integrated surveillance system to monitor AMR bacteria in humans and animals [93], although no evidence to indicate the implementation of such a system has been found. Uruguay listed a series of government institutions and universities that evaluate the incidence of AMR bacteria on its official website but, likewise, no information about a structured program was found [94].

All countries had a system to monitor whether antibiotic residues found in animal foods are within the MRL established by the respective authorities [95,96,97,98,99].

### 2.10. Pharmacovigilance

Only Chile was classified as “strong”, with Brazil, Argentina, and Colombia categorised as “intermediate”. Stakeholders in the pharmacovigilance system are responsible for submitting information about adverse effects to official authorities, and evaluating and applying regulatory measures [100,101,102,103], although the stakeholders involved, their specific responsibilities, and criteria for reporting are different from country to country. Colombia [100] and Chile [101] define the veterinary pharmaceutical industry, veterinarians, farmers, and government authorities as responsible stakeholders, whereas only veterinarians and government officials are mentioned in Argentina [102]. In Brazil, the veterinary pharmaceutical industry and government officials are mentioned [103]. Uruguay holds the veterinary pharmaceutical industry responsible for informing government authorities of any adverse effects on animals and the environment [19], but the government website mentioned that the pharmacovigilance system does not yet include antibiotics [94].

The criteria selected for reporting are also different. Argentina [102], Colombia [100], and Chile [101] included a lack of safety and efficacy, with Argentina also including a lack of quality and stability [102]. Colombia and Chile only specified a lack of safety for animals and humans but also added reports of nonconformity with MRL and harmful effects on the environment [100,102]. Additionally, off-label use was to be reported in Colombia [100], and dissemination of pathogens in Chile [101]. Brazil [103] and Uruguay [19] did not define the criteria for reporting in their legislation.

## 3. Discussion

As a global health problem, AMR bacteria are not contained by geographic frontiers, and no country can tackle this challenge alone [7]. All countries need to implement diverse measures to contain the spread of AMR bacteria. As an important region in the global supply chain of animal foods, South American countries are essential to control the spread of AMR bacteria.

Legislation is one of the main driving forces influencing the amount of antibiotic use in humans and animals [104], and regulatory bodies can use it as a behavioural control tool to influence veterinarians and farmers on antibiotic decision-making. However, national regulatory authorities and other institutions responsible for setting rules in each country are shaped by the socio-historical and economic contexts in these countries, as well as the degree of power and influence that the agricultural sector represents in their societies; in addition, legislation can vary from country to country. However, it is not as simple as enforcing the regulation; it is also necessary to understand gaps in knowledge, culture, and beliefs, as well as the economic, psychological, and political factors behind antibiotic use in humans and animals [104]. Using evidence about these factors, legislative policies can be improved and combined with other strategies to promote behaviour change in the use of antibiotics in livestock production and influence stakeholders (veterinarians, farmers, and animal breeders) to meet antibiotic reduction goals. From our point of view, to ensure food security and public and animal health, and secure space in the global meat trade, South American countries must also strive to think beyond the measures and rules agreed upon and established in international forums.

This paper examined the legislation in five South American countries as scientific evidence of how much progress countries have made in strategies to control AMR bacteria. We mapped some of the points that we consider essential to be observed in all animal-source food production chains, identifying areas where current legislation is stronger, and areas where legislation can be strengthened, needing further development.

### 3.1. Areas with Stronger Legislative Frameworks Points

Marketing authorization is already set in all the evaluated countries, although it is possible to improve some points to reach the gold standard. For example, Colombia could strengthen existing legislation by placing a maximum residue limit as a criterion for antibiotic marketing authorisation. Brazil, Argentina, Uruguay, and Chile could set guidelines or rules on good clinical research practice. These documents can be based on international documents such as the “International Cooperation on Harmonisation of Technical Requirements for Registration of Veterinary Medical Products”. In addition, these countries could establish good laboratory practices regarding antibiotic manufacture, as Colombia and Chile have done. It is important to highlight that all countries have established an official distribution system, with regulations about product labelling and storage conditions. Also, in these countries, only veterinarians can prescribe antibiotics.

Regarding the use of antibiotics as growth promoters, all countries (except for Uruguay and Brazil) prohibit such use of antibiotics, based on the classes established by the WHO as critically important. Uruguay could extend laws to also restrict the use of antibiotic-based growth promoters in the production of pigs and poultry, and Brazil could ban the use of bacitracin and virginiamycin as growth promoters. In addition, all countries could aim to go beyond the ban on the use of growth promoters and oversee that animal feed manufacturers are actually following the legislation and verify that measures are being complied with to prevent cross-contamination between medicated feeds given for different purposes.

Another strong point is that all countries have started to set minimum welfare and biosecurity requirements for food animal producers and are developing similar guidelines about this for farmers. Ultimately, good infection prevention measures could reduce the need for antibiotics, and there is good evidence that the absence of biosecurity measures is associated with higher levels of antibiotic use. Postman et al. [105] have shown that stricter biosecurity measures can reduce the use of antibiotics in swine production, while Albernaz–Gonçalves et al. [106] reported that farmers who did not mention biosecurity measures in interviews used the term “antibiotic shocks” to describe routine preventive strategies to avoid infectious diseases. Linking animal welfare and inappropriate use of antibiotics is more difficult, and there are few studies investigating a direct causal effect. However, a review by Albernaz–Gonçalves [107] presents interesting evidence to start a discussion. The basic idea of the underlying precept is that every stressor can reduce the immune response, becoming a trigger for clinical disease. Stress can be increased by the housing environment, such as weather conditions, high density of animals restricting movement, socialisation, and expression of natural behaviours, and by management practices such as mixing unfamiliar animals, mutilations, prenatal stress, etc. [107] These stressors have also been related to aggressive behaviour (biting) and subsequent serious injuries [108]; infection of piglets during lactation [109]; and urinary and reproductive infections [110]. We recommend that countries continue to develop animal welfare and biosecurity rules and guidelines for different production species. However, setting police restrictions and guidelines may have limited impact [111] and this activity will need to be combined with collective action with all stakeholders (government authorities, agribusiness, and others) to support farmers in improving biosecurity and animal welfare practices [106,107].

With respect to the requirement that farmers keep records of the antibiotics used in animal production, although the OIE gold standard supports this action and some countries have followed it, more research is needed to understand whether this requirement is the best way to measure trends in antibiotic use in animals, or whether farmers are complying with the requirement of keeping records or not.

### 3.2. Areas of Legislation with Room for Improvement

There are currently no regulatory mechanisms regarding the advertising of antibiotics to farmers, or the use of financial incentives to prescribers and suppliers (distributors), in any of the South American countries examined. This is an essential gap that needs to be addressed. A qualitative study with beef cattle producers showed that direct marketing from industry to farmers tends to shape their perception of efficacy and antibiotic choice [112]. Both the European Union [113] and the United Kingdom [114] have already banned this activity, and countries in South America could do the same. Tangcharoensathien et al. [115] reported that healthcare professional prescribing and dispensing can also be influenced by financial incentives, increasing the demand for the unnecessary use of antibiotics. Decoupling financial incentives from the prescription and distribution of veterinary antibiotics can be a tool to promote the rational use of antibiotics [115].

With regard to legislation governing supply chains, just two countries—Colombia and Uruguay—require distributors to keep records of veterinarians’ prescriptions. Brazil and Argentina could follow the same pattern and introduce this policy. Meanwhile, Chile could expand its policy to include other classes of antibiotics beyond fluoroquinolones and cephalosporins. Recently, Chile issued a regulation (Resolucion 4116/2023) [116] establishing an electronic prescription system for antimicrobials. This new regulation will be mandatory from January 2024. We did not include this norm in our analysis since it was created after the period of analysis.

A Brazilian study with pig farmers concluded that free access to antibiotics, without any prescription and sales control, led to reckless use of antibiotics [106], since farmers can buy antibiotics from retail distributors or through industry representatives without a prescription, sometimes even ordering via cell phone messages. The same study showed that farmers mix powdered antibiotics into the feed on their own when they find it necessary. These behaviours do not comply with normative instruction No. 65 [29] and the pig farmers affirmed that they were unaware of this legislation [106]. With an official distribution system and an obligation to keep records, countries can use veterinary prescriptions as data to monitor trends in antibiotic use in livestock and create campaigns for its rational use among all stakeholders [117]. However, it is a critical first step to create regulations and then find a way to influence stakeholders to comply with the rules.

Another area for potential action is storage conditions to preserve the quality of drugs before their use. This is an area where professional associations, together with the veterinary pharmaceutical industry and government authorities, can play a role in creating guidelines on good storage practices. Training the technical managers of the companies responsible for the distribution (wholesalers, retail distributors, animal feed manufacturers) on these topics is a good strategy to ensure the quality and efficacy of the products.

The requirement for a veterinarian to always administer or supervise the administration of antibiotics may not be feasible, given the size of the countries and the number of veterinarians available in each region. When this is not feasible, it is essential that veterinarians clearly describe instructions to farmers or others responsible for the correct administration of the drug (correct dosage and duration).

The importance of guidelines for the responsible and prudent use of antibiotics is fundamental to good antibiotic stewardship. The WHO and OIE lists were created to be used as a tool to preserve the efficacy of antibiotics and can be employed to reduce inappropriate prescriptions of antibiotics. Veterinary professionals and animal feed producer organisations could jointly implement these guidelines, following the priority scale for use set by the WHO [58] and OIE [118] and taking into consideration peculiarities between different animal species and production categories. Our study identified several improvements that are needed in the countries studied. Argentina and Chile currently lack guidelines on the prudent use of antibiotics, and they are urgently needed, while the guidelines in Colombia and Uruguay could be strengthened through the inclusion of guidance on the prioritization of the use of antibiotics and the creation of new guidelines according to different production species. Guidance in Brazil needs to be expanded to include other species (poultry, cattle, etc.).

Regarding therapeutic and metaphylactic use, Argentina, Colombia, and Uruguay could strengthen their disposition towards indicating third- and fourth-generation cephalosporins, fluoroquinolones, and polymyxins only as a last resort; all countries, however, could improve their documents to specifically prohibit the use of these same classes to prevent diseases (prophylaxis). Although Argentina does not allow these antibiotics to be administered via feed, it is still possible to administer antibiotics via other routes such as injections and via water, since there are no regulations available. Concerns regarding these specific classes of antibiotics are based on evidence of high incidence of bacterial resistance to cephalosporins and fluoroquinolones [119], and the discovery in many countries, including Brazil, of resistance genes to colistin, which is one of the last options for multidrug-resistant Gram-negative bacteria in human hospitals [120]. Off-label use remains a major subject of debate. On the one hand, veterinarians have the right to prescribe and take risks, especially where no alternatives are available; but on the other hand, off-label use can contribute to AMR bacteria. The reasons, conditions, and restrictions for off-label use established for Chile are important steps in the right direction. Nonetheless, it is important to investigate if this regulation was sufficient to change the behaviour of prescribers.

Although legislation is one of the driving forces that influence veterinarians when prescribing antibiotics [121], it is advisable to investigate whether legislation with conditions and restrictions would be more effective in each country than a guideline to support a veterinarian’s decision, or whether it would be better to combine both to influence professional behaviour. Furthermore, we must bear in mind that veterinarians as prescribers are not the only agents involved in the decision-making process regarding the use of antibiotics in animals; in multiple ways, farmers as users are also involved, so it is fundamental to change the behaviour of both agents [122]. Knowledge transfer alone is not enough to change stakeholder behaviour [104] and promoting a change in practice faces multiple challenges. Law enforcement can be viewed negatively by farmers and veterinarians as a top-down “overreach” decision [123], which does not correspond to the producer’s reality [124]. It is thus essential to understand the driving forces related to the prescription/use of antibiotics in animals [10]. Some examples described in previous studies include prior experience [125], economic factors [104], workload and time pressure [125], perceived expectations from other partners (veterinarian–client/patient relationship) [126], and deep values that guide attitudes and behaviours [122].

The correct disposal of antibiotics is another gap that needs to be addressed. Accumulation of antibiotic residues in the environment contributes to the problem of AMR bacteria [127]. From our point of view, the correct disposal of unused or expired veterinary antibiotics is essential to mitigate environmental hazards and public health risks such as AMR bacteria. Although it is an improvement when compared to other countries, Colombia and Uruguay only briefly mention that distributors must follow instructions for disposal made by the veterinary pharmaceutical industry, without further instructions. All countries can do more than that and create a reverse logistic policy for veterinary drugs. Reverse logistics for drugs is a social and economic tool that encompasses a series of processes to dispose of expired/unwanted drugs in order to reduce the negative effects on the environment caused by incorrect waste disposal [128]. Brazilian reverse logistics regulation [128] excludes all medicines from animal health services, including zoonosis control centres. Studies have shown that a lack of awareness by society about the environmental and public health damages caused by improper disposal, as well as the lack of funding, are barriers to the implementation of reverse logistics of veterinary drugs [129]. Any change in regulation will therefore also require supportive initiatives to raise awareness and encourage a change in practice.

Surveillance systems and data are crucial supporting infrastructure tools [130] in a One Health approach against AMR bacteria and can be used by experts to monitor the situation, generate actionable knowledge, and influence decision-making to thereby reduce antibiotic consumption and the incidence of AMR bacteria [131]. The method used and the data collected were not sufficient to assess the quality of the monitoring and surveillance systems in each country. Based on the documents evaluated, we can only say which country has a monitoring and surveillance system. However, based on these documents, some recommendations can be made. For example, surveillance data in Uruguay could be strengthened with the inclusion of the class of antibiotics in its sales control system for veterinary products and the creation of a national monitoring and surveillance system for AMR bacteria; Argentina and Colombia could benefit from defining more precisely what type of data they will use in their monitoring and surveillance system. Although data on the consumption and source of antibiotics follow international gold standard recommendations in Brazil and Chile, it would be better if reporting was expanded to include direct sources, such as records from wholesalers and retail distributors (registered prescriptions) or end sources, such as veterinarians and farmers, if this is feasible. Since not every kilogram of antibiotic sold by the veterinary pharmaceutical industry will actually be consumed, collecting sales data from prescriptions or end sources would more accurately express consumption volumes in reality. Chile’s next step is to create an integrated monitoring and surveillance program with both human and animal data about the consumption of antibiotics and AMR Bacteria [93]. This goal matches the literature recommendations to adopt different data and build a range of complex systems to produce strong evaluative evidence [132].

All countries are just beginning to establish their pharmacovigilance system and have not yet defined the scope of the system and the responsibility of each stakeholder (veterinarians, farmers, pharmaceutical industry, and government). Pharmacovigilance systems aim to ensure safety and efficacy to prevent harmful effects on animals, humans, and the environment. The scope of veterinary pharmacovigilance today is quite broad, covering areas such as adverse effects, pharmacological, toxicological, allergic, and microbiological effects of antibiotics. It also covers local reactions at injection sites, lack of efficacy, clinical safety, residues, withdrawal period, and ecotoxicological and environmental issues [133]. In addition, many stakeholders are involved; the veterinary pharmaceutical industry and government authorities are responsible for evaluating reports made by pharmacists, veterinarians, farmers, and animal owners [133]. The Australian Pesticides and Veterinary Medicines Authority (APVMA)—which receives reports from stakeholders and is responsible for performing risk assessments (high, medium, or low) based on evidence, sorting reports into categories (probable, possible, unlikely, or unknown) and then deciding on possible corrective actions, and finally giving feedback to the community [134]—provides a model that could be emulated elsewhere. Countries in South America could continue to develop their surveillance systems, incorporating all areas mentioned in the gold standard, and defining steps to evaluate reports, in addition to specifying the responsibilities of each stakeholder.

## 4. Materials and Methods

This work is based on a document review [135,136] of legislation, guidelines, and official programs currently in force regarding AMR bacteria and the use of antibiotics in livestock production. The documents are available on governmental websites in five South American countries.

### 4.1. Country Selection

As inclusion criteria, we ranked countries according to the quantity of poultry, beef, and pork produced (total amount) and exported (total revenue) and selected the three most important producers and exporters in South America. Additionally, we have taken into account the similarity in patterns of meat consumption, expressed by at least two types of meat consumed. Information about all South American countries is summarised in Table 2. Brazil, Argentina, and Colombia were the leading countries in the production of poultry and cattle. Although Chile does not rank high in these categories, it was the third-largest producer of pork. Also, in terms of exports, Chile was the second largest exporter of poultry and pork, while Uruguay is the third largest exporter of beef. Thus, among the twelve countries in South America, Brazil, Argentina, Colombia, Chile, and Uruguay were selected for analysis of national legislation on the use of antibiotics. All selected countries have at least two of the same type of meat consumption. The five selected countries are classified as high-income countries (HIC) or middle-income countries (MIC) by the World Bank. Legislation data were available on publicly accessible websites for all of them.

### 4.2. Data Sources

Laws, guidelines, and information about official programs were obtained from official websites:Brazil: *Ministério da Agricultura, Pecuária e Abastecimento* (https://www.gov.br/agricultura/pt-br (accessed on 10 March 2022)); *Conselho Federal de Medicina Veterinária* (https://www.cfmv.gov.br/ (accessed on 10 March 2022));Argentina: *Servicio Nacional de Sanidady Calidad Agroalimentaria (*http://www.senasa.gob.ar/normativas (accessed on 10 March 2022)); *Consejo Profesional de Médicos Veterinarios* (https://cpmv.org.ar/) (accessed on 10 March 2022);Colombia: *Ministerio de Agricultura y Desarrollo Rural* (https://www.minagricultura.gov.co/paginas/default.aspx (accessed on 10 March 2022)); *Instituto Colombiano Agropecuario* (ICA) (https://www.ica.gov.co/ (accessed on 10 March 2022)); *Consejo Profesional de Medicina Veterinaria e Zootecnia* (https://consejoprofesionalmvz.gov.co/(accessed on 10 March 2022));Chile: *Servicio Agrícola y Ganadero (*SAG*)* (https://www.sag.gob.cl/(accessed on 10 March 2022)); *Colegio Médico Veterinario de Chile* (https://www.colegioveterinario.cl/public/index.php (accessed on 10 March 2022));Uruguay: *Ministerio da Ganadería, Agricultura y Pesca* (https://www.gub.uy/ministerio-ganaderia-agricultura-pesca/ (accessed on 10 March 2022)); *Colegio Veterinario de Uruguay* (https://www.smvu.com.uy/noticias_379-ley-n-19-258.html (accessed on 10 March 2022)).

Some countries mention in their documents resolutions agreed upon between members of the Mercado Comum do Sul (Mercosul). These regulations, therefore, were also analysed in this paper. The Mercosul resolution was obtained from: https://www.mercosur.int/pt-br/documentos-e-normativa/normativa/ (accessed on 10 March 2022).

The OIE and Codex Alimentarius Standards on the use of antibiotics in animal production and the list of important antibiotics for humans and animals were obtained from https://www.oie.int/en (accessed on 20 December 2021); https://www.who.int/publications/i/item/9789241515528 (accessed on 20 December 2021), and http://www.fao.org (accessed on 20 December 2021).

### 4.3. Data Selection, Analysis, and Synthesis

The selection, analysis, and synthesis of the legislation were guided by predefined categories, using the FAO, WHO [11], and OIE [12] propositions regarding the prudent use of antibiotics in animals as the gold standard. The categories are:Marketing authorization: Process of reviewing and assessing a dossier about an antimicrobial agent to determine whether to allow its commercialization (also called licensing, registration, approval, etc.); when approved, it is finalized with the granting of a document called marketing authorization (equivalent: product license) [11].Container labelling and advertising. Container labelling: All information that appears on any part of a container, including that on any outer packaging such as a cardboard box [138]. Advertising: All informative and persuasive activity by manufacturers and distributors, the effect of which is to induce the prescription, supply, purchase, and/or use of medicinal products. It must be subjected to ethical criteria for drug promotion [139].Distributors: Wholesale or retail distributors of animal antibiotics to animal feed producers.Prescribers: Professionals responsible for prescribing antibiotics to food-producing animals.Therapeutic use: Administration or application of an antibiotic to an individual or groups of animals that show clinical signs of infectious disease [11].Metaphylactic use: Administration or application of antimicrobial agents to a group of animals containing both sick and healthy (presumed to be infected) individuals, to minimize or resolve clinical signs and prevent further spread of the disease [11].Preventive use: Administration or application of antimicrobial agents to an individual or a group of animals at risk of acquiring a specific infection or in a specific situation where infectious disease is likely to occur if the antimicrobial agent is not administered or applied [11].Use for growth promotion: Administration of antimicrobial agents solely to increase the rate of weight gain and/or the efficiency of feed utilization in animals. The term does not apply to the use of antimicrobials for the specific purpose of treating, controlling, or preventing infectious diseases [11].Off-label use: The use of an antimicrobial agent that does not conform to the approved product labelling [11].Producer of animals for food: Worker who exploits the land for economic or subsistence purposes through agriculture, livestock production, and other activities, respecting the social function of the land [140].Monitoring and surveillance programme: A system established to monitor the use of antibiotics in animals (AMU), the incidence of antimicrobial resistance, and antibiotic residues in foods of animal origin [11].Pharmacovigilance: The collection and analysis of data on the performance of products in the field after authorization and any interventions to ensure they remain safe and effective. These data may include information on adverse effects on humans, animals, plants, or the environment, or lack of efficacy [11].

All data (legislation, guidelines, and official programs) were selected considering the FAO, WHO [11], and OIE [12] propositions regarding the prudent use of antibiotics in animals as the gold standard. The following criteria were used:Legislation dealing with authorization of commercialization of drugs.Legislation that addresses container labelling and drug advertising.Legislation that addresses the obligations of wholesalers and retail distributors.Legislation that defines which professionals are responsible for prescribing medications.Legislation, guidelines, and other documents that set rules or recommendations for all purposes of the use of antibiotics.Legislation, guidelines, and other documents that set rules or recommendations for farmers regarding the use of antibiotics.Legislation and websites that prove the existence of a monitoring and surveillance programme.Legislation that set rules on pharmacovigilance.

The levels of development of national legislation on antibiotic use were categorised as strong, intermediate, and weak. For better visualisation, each level was highlighted in different colours (Table 3, Table 4 and Table 5). Definitions at the three levels vary from category to category. In cases where the country’s legislation had better criteria than the gold standard (FAO, WHO, and OIE), the country’s regulation was categorised as the highest level (good).

All Legislation, guidelines, and other documents on veterinary included in this analysis can be access in the Appendix A.

For definition purposes, this paper uses the terms as cited in the OIE Standard [12]. Veterinary medical products (VMPs) containing antibiotics refer to all routes of administration (injectable or oral via feed or water) for different purposes of use (therapeutic, metaphylactic, and preventive); and non-veterinary medical use (growth promoters). 

## 5. Conclusions

The five South American countries examined in this paper are moving towards the international goals set by multilateral agencies such as the WHO, FAO, and OIE. The analysed documents showed strengths and weaknesses. Still, there is a long way ahead. Rules to prohibit advertising to farmers and financial incentives to prescribers are currently lacking and could be developed. There is also room to strengthen existing rules on prescription and use. Some peculiarities and differences between the rules of the countries and the gold standard show a possible adaptation to the realities of these countries. New studies must examine the rationale for these adaptations; whether farmers and prescribers in each country are complying with the legislation that is already in force; and whether the regulations are well suited to the realities faced by farmers. Research is also needed to investigate other tools that can be used alongside legislation as a driving force to change stakeholder behaviour.

## 6. Limitations

As this study was based on publicly available published documents (legislation, guidelines, official websites), it is limited in terms of any possibility of comparing different perspectives on national AMR legislation among each country’s stakeholders or identifying the areas where good consensus or disagreement currently exists. On the one hand, we also recognise that the existence of legislation does not mean it is being implemented, or that it is being complied with by all stakeholders; therefore, the legislation does not necessarily represent each country’s reality. On the other hand, an absence of legislation does not necessarily mean that countries are doing nothing. We must recognise that national action plans (NAPs) on AMR are still being implemented, and some legislation may currently be under development according to a country’s particular needs and challenges and, therefore, may change.

## Figures and Tables

**Figure 1 antibiotics-12-01303-f001:**
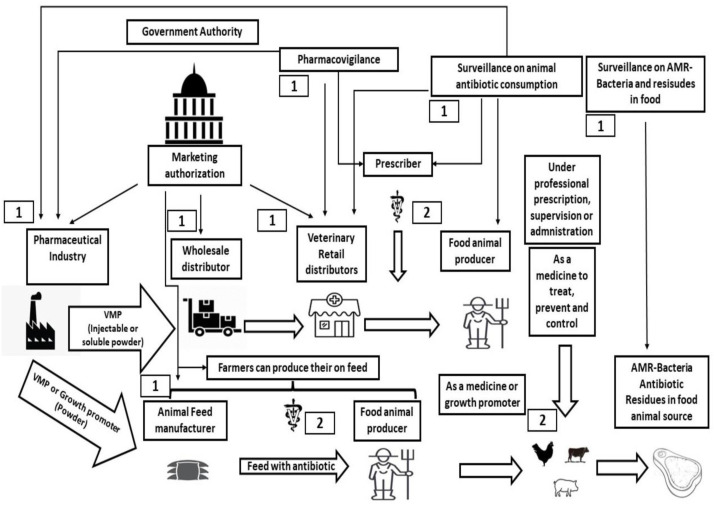
Distribution chain of antibiotics to animals. Source: the authors. Footnote: 1—responsibility of the government to supervise or conduct monitoring and vigilance. 2—responsibility of the veterinarian to supervise.

**Table 1 antibiotics-12-01303-t001:** Level of development of each country’s national legislation on antibiotic use and AMR bacteria.

Categories	Argentina	Brazil	Colombia	Chile	Uruguay
Marketing authorization	Strong	Strong	Strong	Strong	Strong
Container labelling and advertising	Intermediate	Intermediate	Intermediate	Intermediate	Intermediate
Distributors	Intermediate	Intermediate	Strong	Intermediate	Strong
Prescribers	Weak	Strong	Intermediate	Strong	Intermediate
Therapeutic use	Weak	Strong	Intermediate	Strong	Intermediate
Metaphylactic use	Weak	Strong	Intermediate	Strong	Intermediate
Preventive use	Weak	Intermediate	Weak	Intermediate	Weak
Growth promotion use	Strong	Intermediate	Strong	Strong	Intermediate
Off-label use	Intermediate	Intermediate	Intermediate	Strong	Intermediate
Food animal producer	Strong	Intermediate	Strong	Strong	Strong
Monitoring and surveillance	Strong	Strong	Strong	Intermediate	Weak
Pharmacovigilance	Intermediate	Intermediate	Intermediate	Strong	Intermediate

Source: the authors. Colour green: Country has a regulation similar to OIE, FAO, WHO gold standards or proposed an advanced regulation; Colour yellow: Country filled at least half of the criteria established by gold standard; Colour orange: Country filled less than fifty percent of the criteria’s.

**Table 2 antibiotics-12-01303-t002:** Data on meat consumption, export volume, and level of economic development for all South American countries (latest data available).

Country	Type of Meat	Production Per Type of Meat in 2018 ^1^	Exportation (USD) Per Type of Meat in 2018 ^2^	Types of Meat Consumption in 2018	Development Level (2022)
Argentina *	Poultry	2nd (2.11 Mt/year)	3rd (USD 272 M)	1st Poultry	MIC
	Beef	2nd (3.7 Mt/year)	2nd (USD 1.996 B)	2nd Beef	
	Pork	2nd (620.549 t/year)	3rd (USD 21.9 M)	3rd Pork	
Bolivia	Poultry	7th (483.777 t/year)	8th (USD 927k)	1st Poultry	LIC
	Beef	7th (268.000 t/year)	8th (USD 6.39 M)	2nd Beef	
	Pork	9th (109.000 t/year)	No data available	3rd Pork	
Brazil *	Poultry	1st (15.5 Mt/year)	1st (USD 6.06 B)	1st Poultry	MIC
	Beef	1st (9.9 Mt/year)	1st (USD 5.428 B)	2nd Beef	
	Pork	1st (3.79 Mt/year)	1st (USD 1.08 B)	3rd Pork	
Chile *	Poultry	5th (762.318 t/year)	2nd (USD 358 M)	1st Poultry	HIC
	Beef	9th (199.314 t/year)	7th (USD 55.86 M)	2nd Pork	
	Pork	3rd (520.858 t/year)	2nd (USD 452 M)	3rd Beef	
Colombia *	Poultry	3rd (1.59 Mt/year	7th (USD 1.78 M)	1st Poultry	MIC
	Beef	3rd (885 t/year)	6th (USD 71.5 M)	2nd Beef	
	Pork	4th (335.884 t/year)	5th (USD 7.83 M)	3rd Pork	
Ecuador	Poultry	8th (348.836 t/year)	11th (USD 13.7k)	1st Poultry	MIC
	Beef	8th (210.277 t/year)	5th (USD 909.065.2 M)	2nd Beef	
	Pork	5th (249.280 t/year)	9th (USD 6.63 K)	3rd Pork	
Guyana	Poultry	10th (46.322 t/year)	10th (USD 40k)	1st Poultry	MIC
	Beef	11th (2.197 t/year)	No data available	2nd Seafood	
	Pork	12th (629 t/year)	10th (USD 1.34k)	3rd Beef	
Paraguay	Poultry	9th (48.216 t/year)	4th (USD 7.13 M)	1st Pork	MIC
	Beef	5th (495.000 t/year)	4th (USD 1.102 B)	2nd Beef	
	Pork	6th (186.769 t/year)	4th (USD 11,4 M)	3rd Poultry	
Peru	Poultry	4th (1.58 Mt/year)	6th (USD 2.39 M)	1st Seafood	MIC
	Beef	10th (189.703 t/year)	10th (USD 16.1k)	2nd Poultry	
	Pork	8th (162.248 t/year)	7th (USD 119k)	3rd Beef	
Suriname	Poultry	12th (10.877 t/year)	9th (USD 153k)	1st Poultry	MIC
	Beef	12th (1.616 t/year)	9th (USD 72.2k)	2nd Seafood	
	Pork	11th (2.123 t/year)	6th (USD 135k)	3rd Pork	
Uruguay *	Poultry	11th (31.630 t/year)	5th (USD 5.03 M)	1st Beef	HIC
	Beef	4th (589.732 t/year)	3rd (USD 1.643 B)	2nd Seafood	
	Pork	10th (13.175 t/year)	8th (USD 52.7k)	3rd Poultry	
Venezuela	Poultry	6th (665.210 t/year)	No data available	1st Poultry	Did not qualify
	Beef	6th (442.290 t/year)	No data available	2nd Beef	
	Pork	7th (178.804 t/year)	No data available	3rd Seafood	

Source: based on Ritchie and Roser [9], Datawheel [137], and World Bank [138]. Available at: ^1^—https://ourworldindataorg/meat-production (accessed on 10 March 2022); ^2^—https://oec.world/en/home-a (accessed on 10 March 2022); https://datatopics.worldbank.org/(accessed on 10 March 2022). Mt (million tonnes); t (tonnes); LIC (low-income country); MIC (upper middle-income country); HIC (high-income country); M (million); B (billion); K (thousand). * Country selected to the analysis.

**Table 3 antibiotics-12-01303-t003:** Comparative analysis of legislation on antibiotics: categories (m. authorization, container labelling and advertising, and distributors) and levels of development.

Categories	Gold Standard	Strong	Intermediate	Weak
Marketing authorization	Veterinary pharmaceutical industry: Providing all information requested by the competent national authority. Criteria evaluated: quality, safety, efficacy of antibiotics, daily intake (ADI), and maximum residue limit (MRL), all in compliance with the provisions of good manufacturing, laboratory, and clinical practices. Animal feed manufacturer: Implementing appropriate production practices to prevent unnecessary carryover and unsafe cross-contamination of unmedicated feed [12].	The country has a marketing authorization system that required most (more than three) or all criteria demanded in the gold standard in order to register the product and license the manufacturer.	The country has a marketing authorization system but includes less than three criteria established in the gold standard.	The country does not have an official marketing authorization system.
Container labelling and Advertising	Veterinary pharmaceutical industry: The labelling must contain the target animal species, as well as the appropriate age, or production category; therapeutic indications; and withdrawal period. It cannot advertise directly to the animal feed producer and should not provide financial incentives to prescribers or suppliers [12]. Animal feed manufacturer: Ensure appropriate labelling with product identification, withdrawal time, level of medication, approved claim, intended species, instructions for use, warnings, and cautions. It cannot advertise antibiotics directly to the animal feed producer and should not provide incentives that have a financial value to prescribers and suppliers [12].	All information described in the gold standards is included in the labelling. There are also rules to forbid advertising to food animal producers and financial incentives to prescribers or suppliers.	All or most of the information described in the gold standards is included in the labelling, but it does not have rules to forbid advertising to food animal producers or financial incentives to prescribers or suppliers.	There are no established rules on labelling, restrictions on advertising, and financial incentives.
Distributors	Distributor/Retail distributor: Veterinary medical products (VMPs) containing antibiotics must be supplied only through licensed or authorised distribution systems and only upon the prescription of a veterinarian or other person suitably trained and authorised to prescribe according to national legislation. Retail distributors must keep records or copies of prescriptions according to national legislation. Storage and disposal conditions are one of the issues in training on the usage of antibiotics [12]. Animal feed manufacturer: VMPs containing antibiotics and growth promoters must be supplied only through licensed or authorised distribution systems. Manufacturers must provide antibiotic-containing feed to farmers keeping food-producing animals only upon the prescription of a veterinarian or other persons suitably trained and authorised to prescribe in accordance with the national legislation and under the supervision of a veterinarian. The manufacturer must keep records of the feed containing dispensed antibiotics. He must ensure that only approved sources of medications are added to feed at the local level, and for a species and purpose as permitted by the drug premix label or a veterinary prescription. Storage and disposal conditions are one of the issues in training on the usage [12].	There is an authorised distribution system. The provision of VMPs containing antibiotics is carried out upon the prescription of a veterinarian or other authorized professional. Distributors must keep records of the prescription. Only approved sources of medications are added to the feed at the local level and for species and purposes as permitted by the drug premix label or a veterinary prescription. There are guidelines or requirements on storage conditions and/or disposal of the products.	There is an authorised distribution system. The provision is carried out upon the prescription of a veterinarian or other professional. A prescription is required but the distributor does not keep records for all classes of VMPs. Only approved sources of medications are added to the feed at the local level, and for a species and purposes as permitted by the drug premix label or a veterinary prescription. There is a rule, guideline, or requirement on storage conditions and/or disposal of the products.	There is an authorised distribution system, but the provision is carried out without prescription. There is no rule to define approved sources of medications. There are no guidelines, rules, or requirements regarding storage conditions or disposal of the products.

Source: World Health Organization; Food and Agriculture Organization of the United Nations [11]; OIE [12]; World Health Organization [139]. Colour green: Country has a regulation similar to OIE, FAO, WHO gold standards or proposed an advanced regulation; Colour yellow: Country filled at least half of the criteria established by gold standard; Colour orange: Country filled less than fifty percent of the criteria’s.

**Table 4 antibiotics-12-01303-t004:** Comparative analysis of legislation on antibiotics: categories (prescribers, therapeutic, metaphylactic, preventive, growth promotion, and off-label) and levels of development.

Categories	Gold Standard	Strong	Intermediate	Weak
Prescribers	Antibiotics should be prescribed by a veterinarian or other authorised professional based on clinical examination, experience, diagnostic, OIE, and WHO List of Important Antibiotics. They should be administered either under the supervision of a veterinarian or other authorised professional. It is recommended that food animal producer organisations work in cooperation with professional veterinary organisations to implement existing guidelines for responsible, prudent use [12].	The educational qualifications of prescribers are defined. There is at least one guideline for responsible, prudent use (listing all guidelines for prescription), or defined restrictions or conditions for administering antibiotics.	The educational qualifications of prescribers are defined. There is at least one guideline for responsible, prudent use (mentioning most of the guidelines for prescription). No restrictions or conditions for administering antibiotics are defined.	The educational qualifications of prescribers are defined. There are no guidelines. No restrictions or conditions for administering antibiotics are defined.
Therapeutic use	Third- and fourth-generation cephalosporins, colistin, and fluoroquinolones cannot be used as a first-line treatment unless justified, and when used as a second-line treatment, this should ideally be based on the results of bacteriological tests [12].	Guideline(s) or rules take into consideration restrictions mentioned in the WHO AWaRe and OIE lists.	There are guidelines or rules, but they do not take into consideration the restrictions mentioned in the WHO AWaRe and OIE lists.	No guidelines or rules are available.
Metaphylactic use	Third- and fourth-generation cephalosporins, colistin, and fluoroquinolones cannot be used as a first-line treatment unless justified, and when used as a second-line treatment, this should ideally be based on the results of bacteriological tests [12].	Guideline(s) or rules take into consideration restrictions mentioned in the WHO AWaRe and OIE lists.	There are guidelines or rules, but they do not take into consideration the restrictions mentioned in the WHO AWaRe and OIE lists.	No guidelines or rules are available.
Preventive use	Third- and fourth-generation cephalosporins, colistin, and fluoroquinolones should not be used as a preventive treatment administered via feed or water in the absence of clinical signs in the animal(s) to be treated [12].	Guideline(s) or rules take into consideration restrictions mentioned in the WHO AWaRe and OIE lists.	Guidelines or rules indicated preventive use on specific occasions and conditions for antibiotic use but did not mention the WHO AWaRe and OIE lists.	No guidelines or rules are available, or the available guidelines do not mention anything about it.
Growth promotion use	Growth promoters are only authorised for use upon risk analysis. Countries must phase out the authorization of growth promoters based on antimicrobial agents classified as critically important for human health according to the WHO list. E.g.: The third- and fourth-generation cephalosporins, colistin, and fluoroquinolones cannot be used as growth promoters [12].	All antibiotics classified in the WHO Highest Critical Priority Antimicrobials category are prohibited from being used as growth promoters, or all antibiotics are prohibited from being used as growth promoters in all animal production species.	Most antibiotics classified in the WHO Highest Critical Priority Antimicrobials category are prohibited from being used as growth promoters, or growth promoters are banned only for one animal species.	The antibiotics classified in the WHO Highest Critical Priority Antimicrobials category are allowed to be used as growth promoters.
Off-label use	It is allowed under appropriate circumstances and must be in accordance with national legislation. Veterinarians are responsible for defining the conditions of responsible use in such a case. Off-label use should be limited when an appropriate registered product is not available [12].	It is allowed under appropriate circumstances. Veterinarians are responsible for defining the conditions, but the use is limited when an appropriate registered product is not available.	It only mentions that veterinarians are responsible for defining the conditions of responsible use in such a case	Nothing was mentioned about off-label use.

Source: World Health Organization; Food and Agriculture Organization of the United Nations [11]; OIE [12]; World Health Organization [139]. Colour green: Country has a regulation similar to OIE, FAO, WHO gold standards or proposed an advanced regulation; Colour yellow: Country filled at least half of the criteria established by gold standard; Colour orange: Country filled less than fifty percent of the criteria’s.

**Table 5 antibiotics-12-01303-t005:** Comparative analysis of legislation on antibiotics: categories (food animal producer, monitoring and surveillance, and pharmacovigilance) and levels of development.

Categories	Gold Standard	Strong	Intermediate	Weak
Food animal producer	They are responsible for implementing biosecurity measures and animal welfare programmes on their farms to promote animal health and food safety and must keep adequate records of all antibiotics used in animal production [17].	They are responsible for implementing welfare and biosecurity programmes and must keep adequate records.	They are responsible for implementing animal welfare or biosecurity programmes and must (or not) keep adequate records.	They are not responsible for implementing animal welfare and biosecurity program and do not keep records.
Monitoring and surveillance programme	On the use of antibiotics: It varies from country to country. Basic, simple data sources can be used, including import and export data, manufacturing, and sales data; direct sources such as data from registration authorities, wholesalers, retailers, pharmacists, veterinarians, feed stores, and pharmaceutical industry associations; and end-use sources such as veterinarians and food animal producers. On AMR bacteria: It must be scientifically based and may include the following components: 1—statistically based surveys; 2—sampling and testing of food-producing animals on the farm, at live animal markets, or slaughterhouses; 3—organised sentinel programmes, e.g., targeted sampling of food-producing animals, herds, flocks, and vectors; 4—sampling and testing of products of animal origin intended for human consumption; 5—sampling and testing of feed or feed ingredients; 6—assessment of veterinary practice and diagnostic laboratory records. On antibiotic residue: Nothing was found about a surveillance system for antibiotic residue [17].	It has an implemented monitoring and vigilance system on consumption use data from basic, direct, or end sources and an implemented monitoring system of AMR bacteria.	It has an implemented monitoring and vigilance system on consumption use data from basic, direct, or end sources but it does not have an implemented monitoring system of AMR bacteria.	It does not have any monitoring system for antibiotic consumption, AMR bacteria.
Pharmacovigilance	Regulatory authority: It must establish post-marketing antimicrobial surveillance. The information collected through existing pharmacovigilance programmes, including safety, lack of efficacy, and any other relevant scientific data, such as general (animal microorganism) and specific (foodborne and commensal microorganisms) surveillance. Veterinary pharmaceutical industry: It must implement a pharmacovigilance programme and, upon request, provide all the information requested by the competent national authority [17].	The government authority and the veterinary pharmaceutical industry are the responsible stakeholders in the pharmacovigilance system. It includes all criteria evaluated in the gold standard (safety, efficacy, dissemination of microorganisms).	Independent of stakeholders. Not all criteria established in the gold standard are taken into consideration.	It does not have any kind of post-marketing antibiotic surveillance.

Source: World Health Organization; Food and Agriculture Organization of the United Nations [11]; OIE [12]; World Health Organization [139]; Brazil [140]. Colour green: Country has a regulation similar to OIE, FAO, WHO gold standards or proposed an advanced regulation; Colour yellow: Country filled at least half of the criteria established by gold standard; Colour orange: Country filled less than fifty percent of the criteria’s.

## Data Availability

The data presented in this study are openly available on the institutional websites cited in the materials and method section.

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
