# Peer review of "Regulations on the Use of Antibiotics in Livestock Production in South America: A Comparative Literature Analysis"

_antibiotics, 2023, doi:10.3390/antibiotics12081303_

Round 1
Reviewer 1 Report
Review comments for Manuscript ID: antibiotics-2501335
This manuscript entitled “Regulations on the use of antibiotics in livestock production in South
America: A comparative analysis” by Da Silva et al. performed a comparative literature analysis of the regulations governing the use of antibiotics by South American countries in livestock production. The unregulated use and misuse of antibiotics globally have led to the antibiotics resistance crisis the world is currently facing; hence, studies that explore the legislations or regulations that govern antibiotic use are vital, as they could inform stakeholders and policymakers on the menace of this problem. Thus, conceptually, this study is good, well-planned, and executed. That said, there are numerous areas or aspects of the paper that needs thorough revision to enhance the quality of the study and to allow proper comprehension of the study.
First, I suggest a thorough English language review and editing because countless grammar, syntax, word choice/use, sentence linkage, paragraphing, etc., make some parts of the paper difficult to read/understand.
Other specific areas of the manuscript that need attention and revision are as follows:
Title
I think the title should capture the fact that this study is mainly based on literature analysis, so I suggest it be modified to “Regulations on the use of antibiotics in livestock production in South America: A comparative literature analysis”
Abstract
Line 17, all acronyms should be defined on their first usage/appearance.
Line 18, I am not the term “antibiotic medicines” is correct, so please check and revise.
Line 28, I think it would be better to state the gold standard the authors are referring to.
Introduction
Line 35, “an ascendent,” I am unsure if the word “ascendent” is suitable here, so please check and revise.
Line 37, the expression “If neglected action persists” seems to be a tautology, so please check ad revise.
Lines 61-63, the first sentence is ambiguous because what is the subject here? Moreover, the second part could be linked succinctly with the first sentence, or else the 2 are fragmented, which makes it difficult to understand.
Line 64, what is meant by “trade meat partners”?
Line 66, “is South America keeping step?”, is ambiguous, so please check, revise, and combine succinctly with the preceding sentence.
Lines 67-70, this is a long and convoluted sentence, please break down and revise clearly.
Results
Box 1, can authors include the definitions for ‘strong’, ‘intermediate’, and ‘weak’, in the footnotes.
Line 142, I suggest you use a good linking word or phrase to link this part with the previous paragraph, or better still, combine this short paragraph with the preceding one.
Line 151, I think a reference is needed here on the gold standard.
Lines 163-164, please combine this short paragraph with the preceding or the next one.
Lines 169-172, combine this paragraph with the next one.
Line 181, I think the word ‘prescribe’ should be ‘prescription’
Line 182, what is meant by the expression “escalation of use”?
Lines 184-189, revise this paragraph succinctly and combine it with the preceding paragraph.
Line 192, see comment on the expression “escalation of use”
Lines 197-199, combine this paragraph with the preceding one.
Line 213, “for all finalities of use”, This is an ambiguous phrase, please check and revise.
Line 219, please be specific, "as above" here means where? The preceding sentence, paragraph or?
Line 224, “finality”, what is the meaning of this term/word that has been used? please use a better alternative.
Lines 223-226 and lines 227-236, combine these paragraphs and link succinctly with a proper linking word/phrase.
Lines 260 and 262, please define all acronyms on their first use.
Lines 269-271, please combine with the preceding paragraph, summarize, and revise contently succinctly.
Lines 294-295, combine with the preceding paragraph and link succinctly.
Lines 297-313, combine these 2 paragraphs. Also, delete the sentence “No results were published yet.” or link it with others, else it is ambiguous.
Lines 305-308, these 2 sentences are a bit fragmented and should be combined with the preceding one and linked succinctly.
Line 312, define all abbreviations or acronyms on their first usage.
Discussion
Line 380, delete the word “same”
Line 384, change “of them” to "these countries".
Line 397, change “has” to "have".
Line 528, delete “should”
Materials and methods
Line 563, what were the criteria used for the selection of the documents to review? This needs to be mentioned and supported by evidence.
Line 579, define all abbreviations or acronyms on their first appearance.
Other minor comments
I am not too sure why the manuscript was arranged in this order, i.e., after the discussion came the materials and methods, then followed by the conclusion and limitations of the study. please check and make sure the arrangement is in line with the journal's requirements.
I suggest a thorough English language review and editing because countless grammar, syntax, word choice/use, sentence linkage, paragraphing, etc., make some parts of the paper difficult to read/understand.
Author Response
Dear revisor I,
Thanks for your suggestions.
I answered everything in the document attached

Reviewer 2 Report
SUMMARY
The authors compared publicly accessible documents which detail Antimicrobial resistance-related legislation in 5 major South American countries. The authors clearly state the limitations of the study, for example, the existence of legislation does not mean that it is being implemented. Gaps were identified and suggestions were advanced with respect to improvements that can be made. Also, highlighted were the complexity of behavior change and the need for further study in various areas.
GENERAL comments
- The word "Box" should be replaced with "Table" for Tables 1 to 5
- The numbering of pages after Page 15 should be corrected
- The title of the WHO document 61-2005 is "Code of Practice to Minimize and Contain Foodborne Antimicrobial Resistance". The word "Foodborne" was omitted in the title of Reference number 11.
SPECIFIC comments
Page 8 line 368 "four" should be replaced with "five"
The article should be checked for some grammatical errors.
Author Response
Dear reviewer 2
Thank you for your suggestions,
I answered everything in the document attached.

Round 2
Reviewer 1 Report
Most of the issues raised have been dealt with by the authors